# Can Perceivers Differentiate Intense Facial Expressions? Eye Movement Patterns

**DOI:** 10.3390/bs14030185

**Published:** 2024-02-26

**Authors:** Leyu Huang, Tongtong Zhu, Jiaotao Cai, Yan Sun, Yanmei Wang

**Affiliations:** 1School of Psychology and Cognitive Science, Shanghai Key Laboratory of Mental Health and Psychological Crisis Intervention, East China Normal University, Shanghai 200062, China; 2Shanghai Changning Mental Health Center, Shanghai 200335, China

**Keywords:** intense facial expression, winning faces, losing faces, forced-choice response task, eye tracking

## Abstract

Recent research on intense real-life faces has shown that although there was an objective difference in facial activities between intense winning faces and losing faces, viewers failed to differentiate the valence of such expressions. In the present study, we explored whether participants could perceive the difference between intense positive facial expressions and intense negative facial expressions in a forced-choice response task using eye-tracking techniques. Behavioral results showed that the recognition accuracy rate for intense facial expressions was significantly above the chance level. For eye-movement patterns, the results indicated that participants gazed more and longer toward the upper facial region (eyes) than the lower region (mouth) for intense losing faces. However, the gaze patterns were reversed for intense winning faces. The eye movement pattern for successful differentiation trials did not differ from failed differentiation trials. These findings provided preliminary evidence that viewers can utilize intense facial expression information and perceive the difference between intense winning faces and intense losing faces produced by tennis players in a forced-choice response task.

## 1. Introduction

Understanding others’ facial expressions is essential to social interaction; successful social interactions require an individual to accurately perceive, recognize, and comprehend others’ facial expressions. Most previous studies on facial emotion perception of moderate intensity have revealed that both static and dynamic expressions are highly recognizable in healthy adults [1,2,3,4,5,6]. Few studies have been conducted to investigate facial emotion recognition using spontaneously intense emotional expression [7,8]. It remains unclear how distinct peak intensity expressions of opposite affective valence are.

Intense emotion (i.e., peak emotion) refers to highly intense emotions occurring in real life with a focus on transient peak expressions, such as the triumph of victory, the grimace of pain, etc. [7,8]. Aviezer et al. (2012) used photographs of tennis players depicting reactions to losing or winning a point in a high-stakes tennis competition as experimental materials and revealed that participants could not differentiate winning faces from losing faces during intense situations. Specifically, both intense winning faces and intense losing faces were rated as equally negative. Researchers proposed that during peak intensity moments, the facial muscles are extremely constrained, and the configuration distortion caused by high intensity makes the peak facial expressions hard to distinguish. However, another study found that intense winning faces and intense losing faces are physically distinguishable [7]. Specifically, the researchers found that intense winning faces contained more facial activity involving smiling (AU12), eye constriction (AU6), and mouth opening (AU27) than intense losing faces, indicating an objective difference in facial activity between intense winning faces and intense losing faces. While objective differences between intense winning and losing faces are detectable, participants failed to judge the valence of intense real-life emotional expressions using valence-rating tasks [7].

In these studies, researchers have adopted a classical valence-rating paradigm in which participants were required to judge the valence of an isolated image (winning face or losing face) in each trial [7,8]. This experimental manipulation might be too difficult for perceivers to perceive intense winning/losing faces. Therefore, in the present study, we developed a novel experimental task (forced-choice response task) in which a pair of images of tennis players’ emotional faces (an intense winning face and an intense losing face) were displayed to participants simultaneously, and participants were asked to select the winning face or losing face from a pair of intense facial expressions. Furthermore, during high-stakes tennis competitions in real life, the audience usually observes winning faces and losing faces produced by the two professional tennis rivals simultaneously. Therefore, a forced-choice task might be more helpful in increasing ecological validity relative to the valence rating frequently used in previous studies [7,8]. In addition, previous studies have also proposed that a forced-choice response to the emotional expression paradigm is a simple, clear, and methodologically strong technique, and it is a good option when the ability to discriminate between positive emotions and negative emotions is of interest [9]. Therefore, in the present study, we adopted a forced-choice response task to explore whether perceivers could distinguish the valence of real-life intense facial expressions.

### 1.1. Related Work

The ability of facial recognition is helpful to mental health and human development [10,11,12]. However, previous work has mainly focused on the recognition of posed facial expressions. Posed expressions are used as a means of faking, masking, or suppressing emotional experiences that have been reproduced in an experimental environment and displayed or imitated by actors voluntarily [13,14,15]. These results have demonstrated that posed facial expressions of moderate intensity convey positive and negative valence in a highly distinct manner, and there are objective differences between moderate positive expressions and negative expressions in facial activities [16,17].

Research has also explored the associations between action units (AUs) and the valence of facial expressions and revealed that two facial action units, namely, brow lowering (AU4) and smiling (AU12), might be direct indices of valence. Positive facial expressions are associated with more zygomatic facial muscle activity than negative facial expressions, and negative facial expressions (such as sad and angry expressions) are associated with higher corrugator muscle activity [17,18,19]. A smiling mouth (AU12) has been found to enhance perceived pleasantness and contribute more to happiness than positive eyes [20,21,22,23]. The activation of brow lowering (AU4) is more often associated with negative expressions [15,24,25]. More importantly, research has also revealed that the smiling mouth (AU12) is related to positive emotions that are high on arousal and power, whereas brow lowering (AU4) is associated with negative, high-arousal, and low-power emotions [21].

Since posed expressions idealize and often exaggerate nature, posed expressions may be unrepresentative of spontaneous affective expressions commonplace in everyday life [26,27,28,29,30]. Therefore, the ecological validity of research using posed facial expressions may be compromised. In the present study, we focused on the recognition of spontaneous facial expressions. Since we focused on the recognition of intense winning faces (high arousal in positive emotion), intense losing faces (high arousal in negative emotion), we speculated that a smiling mouth (the lower part of the face) is more likely to be related to intense winning faces, and brow lowering (the upper part of the face) is more likely to be related to intense losing faces. In addition, we also adopted moderate facial expressions spontaneously (moderate winning faces/moderate losing faces) in the present study to compare the recognition performance difference between intense and moderate winning faces/losing faces using a forced-choice response paradigm.

Gaze direction and attention focus are assumed to be tightly coupled; therefore, eye-tracking technology is one suitable method to directly reflect an individual’s immediate processing regarding facial expression recognition [31,32,33]. Additionally, another advantage of measuring eye movement is that it is conducted in real time and noninterfering, meaning the experiment can feel more natural for participants [10]. Recently, there has been a growing body of research examining gaze behavior during the perception of emotional faces [34,35,36,37]. Evidence shows that distinguishable gaze patterns exist for positive expressions and negative expressions. Specifically, participants have exhibited longer fixation durations and initially fixated on the eyes in negative facial expressions (sad faces, fear faces, angry faces) relative to happy facial expressions [31,38,39].

### 1.2. The Current Research

Given previous findings, using eye-tracking techniques, the present study aims to explore whether perceivers can perceive the objective difference between intense positive expressions and intense negative expressions and judge the valence of intense facial expressions above chance level in a forced-choice response task. We hypothesized that participants could distinguish intense winning faces from intense losing faces produced by professional athletes in a forced-choice response task above chance, which was reflected by recognition accuracy and gaze patterns. Specifically, for recognition accuracy, we expected that recognition accuracies for both intense winning faces and intense losing faces would be higher than the chance level. For eye movement patterns, we also hypothesized that participants exhibit differential eye movement patterns for intense winning faces and intense losing faces. Specifically, participants would be more likely to fixate on the lower part of an intense winning face and the upper part of an intense losing face. In addition, we also adopted real-life emotional expressions of moderate intensity produced by professional tennis players as the control condition for intense emotional expressions.

## 2. Method

### 2.1. Participants

According to previous studies on facial expression recognition using the brief fixation paradigm [28,34,40], which ranged from 24 participants [41] to 32 participants [40] for within-participants comparisons, we aimed for a sample size larger than those of most previous studies. Based on the effect size reported in these previous studies, we conducted an a priori power analysis using G*Power 3.1 [42] for a repeated measurement ANOVA to detect an effect size of *f* = 0.20 with a statistical power of 0.95 and a significance level of 0.05; we aimed to recruit a sample size that included a minimum of 36 participants. In total, 36 undergraduate students (24 female, 22.92 ± 1.12 years, range: 21–25 years, Chinese ethnicity) participated in this experiment. All participants were right-handed and had either normal or corrected-to-normal vision. None of the participants were aware of the purpose of the experiment. The present study was approved by the local research ethics committee. All participants provided written informed consent before the experiment. Upon completion of the study, participants were paid for their participation.

### 2.2. Materials

Intense facial expressions. Images of tennis athletes either winning or losing a point in a professional high-stakes tennis match, which would typically evoke strong affective reactions, were obtained from Google Image Search using the query “reacting to winning a point” or “reacting to losing a point” crossed with “tennis”. Images were searched using the same keywords as [7,8]. We selected 25 intense winning faces and 25 intense losing faces. We mainly focused on these facial action units: brow lowering (AU4), smiling (AU12), constriction (AU6), and mouth opening (AU27) [8]. Each intense face contained at least three action units of these action units. The following selection criteria were used to ensure that the selection process was random: (1) Gender distribution in the images was nearly equal: thirteen of the losing images and fourteen of the winning images showed males. (2) In total, 21 of the 25 losing faces s and 21 of the 25 winning faces were unique identities. (3) The face was cropped from each image and enlarged. (4) All images were converted into black and white. (5) All pictures were presented in grayscale, with a gray background, and were normalized in terms of resolution, contrast, and luminance. (6) The orientations of the intense facial images were forward, 50%; right, 45%; and left, 5%. The orientation of the facial images was not associated with winning or losing a point; $x^{2}$(2) = 2.02, *p* = 0.45.

Moderate facial expressions. Moderate facial expressions were selected from tennis match videos within 1000 ms after the appearance of an intense emotional face [43]. Moreover, according to previous research on specific morphological facial features (action units; AUs) related to positive emotions and negative emotions [8,40,41,44], if a facial expression included one of four obvious facial movements that indicated positive emotion or negative emotion, we defined it as a moderate emotional expression. Using this operational definition, we captured critical facial expressions from professional tennis match videos, and the poses of the critical facial expressions were converted from video into still images. Then, we acquired 25 moderate winning faces and 25 moderate losing faces containing typical facial movements associated with positive and negative expressions. Thirteen of the losing images and thirteen of the winning images showed males. In total, 22 of 25 losing faces and 21 of 25 winning faces were unique identities. The orientation of the moderate facial images was forward, 36%; right, 36%; and left, 28%.

A total of 100 facial pictures comprised the experimental materials. All pictures were edited using Photoshop CS6 to ensure uniformity of luminance and facial size. All pictures had a display size of 5 cm × 5 cm and a resolution of 533 × 350 pixels with an average visual angle of 4.8°. Isolated faces were displayed in grayscale against a gray background.

#### 2.2.1. Discrimination of Intense Expressions and Moderate Expressions Check

To confirm that there is a difference in the valence/arousal of these two categories of isolated faces, we recruited 45 college students (22.45 ± 3.52 years, range: 21~25 years, 34 female) to rate the valence (1 = most negative; 7 = most positive) and the arousal (1 = very calm, 7 = very excited) of these expressions on a 7-Likert point scale [45]. Each image was presented separately. In the arousal discrimination block including 100 trials (50 intense faces, 50 moderate faces), participants rated the arousal of images using a 7-Likert point scale ranging from 1 (very calm) to 7 (very excited). In the valence discrimination block containing 100 trials (50 intense expressions, 50 moderate expressions), participants rated the valence of images using a 7-Likert point scale ranging from 1 (most negative) to 7 (most positive) [45]. At the beginning of each trial, participants were instructed to look at the fixation point at the center of the screen lasting 1000 ms. Subsequently, an isolated face image was randomly presented at the center of the screen. The image was presented for unlimited duration until a response was carried out. Each face was presented twice (once in the valence-rating block and once in the arousal-rating block).

#### 2.2.2. Valence Ratings of Isolated Intense Faces and Moderate Faces

For intense faces, a 2 (intensity: intense vs. moderate) × 2 (face type: winning faces vs. losing faces) repeated-measures ANOVA was conducted on the mean valence score. The main effect of intensity was not significant (*F* (1, 44) = 1.83, *p* = 0.19 > 0.05), indicating that there was no significant difference between the valence score of intense expressions (3.05 ± 1.20) and that of moderate expressions (3.31 ± 0.65). The main effect of face type was significant (*F* (1, 44) = 61.28, *p* < 0.001, $\eta_{p}^{2}$ = 0.69), indicating that the valence score of winning faces (3.70 ± 1.03) was significantly higher than that of losing faces (2.67 ± 0.82). The interaction between intensity and face type was also significant: *F* (1, 44) = 30.26, *p* < 0.001,${\;\eta }_{p}^{2}$ = 0.52. A further simple effect analysis indicated that valence score between intense winning (3.06 ± 1.32) and intense losing faces (2.94 ± 0.99) showed no significant difference:

*F* (1, 44) = 0.48, *p* = 0.49 > 0.05; however, the valence of moderate winning faces (4.54 ± 0.65) was significantly higher than moderate losing faces (2.08 ± 0.65), *F* (1, 44) = 57.69, *p* < 0.001.

#### 2.2.3. Arousal Ratings of Isolated Intense Faces and Moderate Faces

A 2 (intensity: intense vs. moderate) × 2 (face type: winning faces vs. losing faces) repeated-measure ANOVA was conducted on the mean arousal score. A significant main effect of intensity was found (*F* (1, 44) = 70.89, *p* < 0.001, $\eta_{p}^{2}$ = 0.71), indicating that intense faces (5.88 ± 1.23) were rated as higher arousal than moderate faces (3.42 ± 0.83). Specifically, intense winning faces (5.94 ± 1.20) were rated as higher arousal than moderate winning faces (3.56 ± 0.89): *t* (44) = 6.74, *p* < 0.001. Similarly, higher arousal was reported for intense losing faces (5.85 ± 1.26) relative to moderate losing faces (3.28 ± 0.78): *t* (44) = 10.23, *p* < 0.001. The main effect of face type was not significant: *F* (1, 44) = 0.76, *p* = 0.39 > 0.05. Further analysis revealed that the arousal of intense winning faces (5.90 ± 1.12) did not differ from that of intense losing faces (5.86 ± 1.27). There was no significant difference in arousal scores between moderate winning faces (3.55 ± 0.89) and moderate losing faces (3.29 ± 0.79). There was no significant interaction between intensity and face type: *F* (1, 44) = 2.01, *p* = 0.18 > 0.05.

#### 2.2.4. Experimental Task and Procedure

Same-intensity winning and losing images were paired (i.e., a pair of an intense winning face and an intense losing face or a pair of a moderate winning face and a moderate losing face). In each trial, paired images of tennis players’ emotional faces were displayed to participants simultaneously, depicting a winning face and a losing face. The present study consisted of five blocks (one practice block and four experimental blocks). Experimental blocks included an “intense winning emotion task” block, a ”moderate winning emotion task” block, an “intense losing emotion task” block, and a ”moderate losing emotion task” block. In the “intense winning emotion task” block, participants were asked to select the winning face from a pair of intensely emotional faces. In the “intense losing emotion task” block, participants were asked to choose the losing face from a pair of intensely emotional faces. Similarly, participants were required to choose the corresponding facial expression according to experimental instructions for moderate emotional expressions (Figure 1A). Participants could only choose one face from a pair of images; therefore, we adopted a forced-choice task. Each trial began with a picture of experimental instructions that occurred 500 ms and then a central fixation cross lasting 500 ms. After the offset of fixation, a pair of images was presented until participants made a response. Participants were required to press the “F” or “J” key if the target face (the winning face or losing face according to experimental instructions) appeared on the left or right, respectively. The position (i.e., the left or right side) of the target face was randomly selected by the computer program. To avoid stimulus–response compatibility effects, the response key assignment was switched in the middle of the experiment. After participants made a response, there followed an inter-trial interval of 1000 ms.

The practice block consisted of 12 trials, including 3 trials of the “winning intense emotion task”, 3 trials of the “moderate winning emotion task”, 3 trials of the “intense losing emotion task”, and 3 trials of the “moderate losing emotion task”, after which, participants entered the main experimental blocks. Each experimental block included 50 trials, each paired with same-intensity winning and losing images repeated once. The order of the four experimental blocks was presented randomly for each participant, and individuals were given unrestricted time to respond to ensure sufficient time to observe two facial expressions and make judgments. Two block tasks were separated by 5 min intervals. The order of experimental blocks was counterbalanced between participants.

### 2.3. Apparatus

Eye movements were recorded using an Eyelink1000 eye-tracker from SR Research at a sampling rate of 1000 Hz. The eye-tracker was calibrated to each participant’s dominant eye, but viewing was binocular. Before experimental trials, the tracker was calibrated for each participant using a 9-point automated calibration accuracy test. In this procedure, the tracker adjusts and recalibrates until tracking error values reach 0.5° of the visual angles for the *x*- and *y*-axes. Calibration was repeated if the error exceeded 18 at any point or if the average of all points was greater than 0.58. During experimental trials, the tracking device continuously stored coordinates of participants’ gaze positions on the screen at a frequency of 1000 Hz. Eye movement data with signal loss due to blinking or off-screen gazes were excluded automatically. Participants placed their heads on a chin rest for better positioning. The computer screen was approximately 70 cm away from their eyes, and the display resolution was set to 1024 × 768 pixels. Face images were presented with a size of 4.5 × 4.5 cm (400 pixels) with horizontal and vertical angles below 5°.

### 2.4. Eye-Tracking Data Processing

Before the start of the experiment, each participant completed a standardized 9-point calibration and validation procedure. At the beginning of each presentation block, an additional 1-point calibration was completed to ensure a constantly high data quality. After completion of the experiment, the raw eye-tracking data were preprocessed in Matlab version 2016b [46]. Trials were excluded from data analysis if one or more of the following exclusion criteria were met: (a) there was no fixation data for 25% or more of the trial duration; (b) 50% or more of the data points during picture presentation were interpolated [47,48]; (c) participants did not fixate on the fixation cross before the picture onset, as defined by a deviation in the average gaze by more than ± 3 SDs of the mean during that period.

## 3. Results

### 3.1. Behavioural Results

Accuracy. To examine whether the recognition accuracy for moderate faces was higher than for intense faces, a 2 (intensity: moderate vs. intense) × 2 (task type: choosing winning face vs. choosing losing face) repeated-measures analysis of variance (ANOVA) was conducted to assess the judgment accuracy rating. A significant main effect was found for facial expression intensity (*F* (1, 35) = 74.18, *p* < 0.001, $\eta_{p}^{2}$ = 0.69), indicating that participants judged moderate facial expressions more accurately (0.83 ± 0.11) compared with intense emotional faces (0.63 ± 0.17).

A chi-squared test was conducted to test whether participants could recognize the valence of emotional faces in a forced-choice task. For intense faces, a chi-squared test revealed considerable differences in the intense winning emotion task: χ^2^ = 127.89, *p* < 0.001. Participants selected intense winning faces (total = 50 trials, *M* = 31.14, *SD* = 3.97) more often than they selected intense losing faces (total = 50 trials, *M* = 18.86, *SD* = 3.97) when participants were required to select the winning face from a pair of intense faces. There was a significant difference in the selecting frequencies when participants were asked to choose intense losing faces from a pair of intense faces: χ^2^ = 135.31. *p* < 0.001. Participants selected intense losing faces (total = 50 trials, *M* = 31.54, *SD* = 4.76) more often than they selected intense winning faces (total = 50 trials, *M* = 18.46, *SD* = 4.71).

According to previous studies on the statistical methods of comparing means estimation with chance [49,50,51], we also carried out two separate one-sample *t*-test analyses to examine whether the recognition accuracy of intensely emotional faces exceeded the chance level (50%). The accuracy of the intense losing faces was 0.632 ± 0.18, *t* (35) = 4.16, *p* < 0.001, Cohen’s *d* = 0.70, and the accuracy of the intense winning faces was 0.623 ± 0.16, *t* (35) = 4.57, *p* < 0.001, Cohen’s *d* = 0.71. These results suggest that the recognition accuracy of the valence of intensely emotional faces was significantly above chance (Figure 1C).

For moderate faces, participants selected moderate winning faces (total = 50 trials, *M* = 40.34, *SD* = 2.35) more often than moderate losing faces (total = 50 trials, *M* = 9.08, *SD* = 1.88) when participants were required to select the winning face from a pair of moderate faces, χ^2^ = 82.56, *p* < 0.001. Participants selected moderate losing face (total = 50 trials, *M* = 42.28, *SD* = 3.19) more often than moderate winning faces (total = 50 trials, *M* = 7.72, *SD* = 3.19) when participants were required to select the losing face from a pair of moderate faces: χ^2^ = 108.5, *p* < 0.001. These results suggest that participants could recognize the valence of the moderate emotional faces successfully.

### 3.2. Eye-Movement Results

We defined two regions of interest (ROIs): (a) the upper part of the face (eye region), encompassing the left and right eyes, including the eyebrows and the nasion area in-between the eyes, and (b) the lower part of the face (mouth region), consisting of the partial body of the nose (tip-defining points, alar-sidewall, and supra-alar crease) or the “mouth” and immediate surrounding area (left and right cheeks). The dividing line between the upper and lower parts was a horizontal line along the tip of the nose [52]. The ROIs were defined as generously sized rectangles around the facial features; the face regions’ shape was fixed relative to the face but changed according to the facial orientation; the regions of interest containing the two eyes were always visible. Fixations were defined as any period that was not a blink or saccade (with default thresholds automatically identified by the eye-tracking system) and lasted at least 100 ms (Eyelink Dataviewer User’s Manual, 2002–2008, SR Research Ltd., Missis-sauga, Ontario, Canada). Two eye movement indices were considered: fixation counts (mean number of fixations on ROIs for each face) and fixation duration (total fixation dwell time on ROIs for each face).

#### Fixation Counts on Emotional Faces

Then, we investigated whether participants perceived the physical difference between winning faces and losing faces, which were reflected on the eye-tracking index, and we also examined whether the gaze patterns on two regions of interest for intense winning faces were different from those for intense losing faces. A 2 (intensity: intense faces vs. moderate faces) × 2 (ROI: upper region vs. lower region) × 2 (face type: winning faces vs. losing faces) × 2 (task type: choosing the winning face vs. choosing the losing face) repeated-measures ANOVA on fixation counts revealed the significant main effect of intensity (*F* (1, 35) = 24.26, *p* < 0.001,$\;\eta_{p}^{2}$ = 0.46), with more fixation counts on intense faces (2.67 ± 2.32) than those on moderate faces (1.78 ± 1.46). The main effect of ROI was significant (*F* (1, 35) = 5.36, *p* < 0.05,$\;\eta_{p}^{2}\;$= 0.16), indicating that participants were more likely to fixate on the eye region (2.83 ± 2.38) than on the mouth region (1.62 ± 1.39). More importantly, there was a significant three-way interaction (intensity × ROI × face type): *F* (1, 35) = 12.30, *p* < 0.01, $\eta_{p}^{2}$ = 0.31. To clarify this significant three-way interaction effect to examine fixation count differences in the ROI regions between recognizing winning faces and recognizing losing faces, we conducted two separate ANOVA analyses for intense faces and moderate faces.

First, for average fixation counts on intense faces, there was a significant two-way interaction between ROI and face type (*F* (1, 35) = 58.03, *p* < 0.001, $\eta_{p}^{2}$ = 0.63), revealing that participants were more likely to fixate on the lower facial region (mouth) of intense winning faces (2.89 ± 1.78) than intense losing faces (1.38 ± 1.12). However, the findings revealed more fixation counts on the upper facial region of intense losing faces (3.93 ± 2.79) than those of intense winning faces (2.26 ± 2.39) (Figure 2A).

Second, for average fixation counts on moderate faces, the results showed a significant interaction between face type and ROI (*F* (1, 35) = 63.91, *p* < 0.001, $\eta_{p}^{2}$ = 0.65), indicating that participants fixated more on the eye region of moderate losing faces (2.63 ± 1.32) compared with moderate winning faces (1.58 ± 1.55). Conversely, participants fixated more on the mouth region of moderate winning faces (2.01 ± 1.19) relative to moderate losing faces (0.78 ± 0.93) (Figure 2C).

The interaction of face type × task type was also significant (*F* (1, 35) = 63.41, *p* < 0.001, $\eta_{p}^{2}$ = 0.69), indicating that participants fixated more on losing faces (2.67 ± 1.17) than on winning faces (2.23 ± 1.16) in the losing task condition. By contrast, participants were more likely to fixate on winning faces (2.22 ± 1.04) than on losing faces (1.78 ± 0.87) in selecting the winning face task condition. No other significant effect was found: *F*s < 2, *p* > *0*.05.

We also compared the difference in eye movements on intense faces between successful trials of differentiation for intense faces from winning faces and failed trials of differentiation. Regarding the winning intense emotion task type (selecting the winning face from a pair of intense faces), there was no significant difference in the average proportions of fixation on the face between the trials of successful differentiation and the trials of failed differentiation, regardless of face type (winning faces vs. losing faces): *ts* < 1, *p* > 0.05. For the intense losing task type (selecting the losing face from a pair of intense faces), a similar result pattern was also observed.

## 4. Discussion

The present study adopted a forced-choice paradigm and investigated whether participants might perceive the difference between intense positive faces from intense negative faces during peak intensities of emotion in professional tennis matches. The results revealed that the accuracy of valence judgment for these facial expressions significantly exceeded chance (50%): 63.20% and 62.29% for intense losing and intense winning faces, respectively. Regarding eye movement patterns, similar to moderate facial expressions, participants were more likely to fixate on the upper facial region (eyes region) than the lower facial region (mouth region) for intense losing faces. Conversely, participants exhibited more fixation counts on the lower part of the face (mouth region) than the upper part of the face (eye region) for intense winning faces. These results provided preliminary evidence that participants might become aware of the objective difference between intense positive and negative faces by utilizing information in the facial movements of victorious and defeated faces.

Research has revealed that intense facial expressions apparently fail to transmit emotional valence information when isolated intense facial expressions are presented sequentially; participants rate both categories as indicative of negative valence [7,53,54]. Scholars have also found objective differences in facial muscular activity between intense winning faces and intense losing faces [8], such that intense winning faces demonstrate more facial muscle activities, especially smiling (AU12) and mouth opening (AU27), compared with intense losing faces. In these studies, participants were asked to rate the valence of the target’s facial expression using a single Likert scale [7,8,53,55]. We adopted a forced-choice task in which intense winning and intense losing faces were displayed simultaneously to compare objective differences in facial movements. In our study, participants were asked to choose the winning face or the losing face from a pair of intense faces, which could provide more context information to make a response for participants. Furthermore, participants had unlimited time to respond and made judgments in a two-way forced-choice situation. Therefore, the experimental task in the present study was different from the valence-rating task, which has mostly been used in prior research [7,8,53,55]. A lower accuracy rate in recognizing intense facial expressions might reflect a transient signal at the facial muscular level [8]. We could not conclude that intense facial expressions did not lose their function of conveying emotional information. The present study provides original evidence that participants performed better on the forced-choice intense face recognition task than on the valence-rating judgment task of a single intense face. Considering that our experimental manipulation was more similar to the situation of a professional tennis match in real life, where audiences can simultaneously observe the emotional faces of two tennis players in a professional tennis match, we speculated that the findings of the present study might be helpful in improving ecological validity.

As predicted, the participants recognized moderate facial expressions more accurately than intense facial expressions. Moderate facial expressions in the present study were created by previous studies based on the association between facial action units and facial expressions, similar to posed facial expressions, which have standardized stereotypical configurations of facial muscle movements to signal positive emotion or negative emotion [56,57]. In the present study, moderate facial expressions also contain critical facial action units that could convey information about facial expression valence in both positive and negative contexts [8,57,58]. Therefore, we also demonstrated that perceivers could recognize moderate emotional expression successfully. These results for moderate emotional expressions align with previous results indicating a recognition advantage for posed prototypical facial expressions that are produced with a characteristic configuration of facial muscle movements [59,60]. In addition, the behavioral results revealed that participants chose the moderate winning face more quickly than choosing the moderate losing face from a pair of moderate faces, further confirming a recognition advantage for positive faces in previous studies found on posed expressions [59,61,62,63]. Finally, contrary to previous findings showing that high intensity can facilitate facial expression recognition [64], our findings suggest that the recognition of extremely intense facial expressions is worse than that of moderate-intensity expressions. This pattern raises the possibility of an inverted U-shaped curve for expression recognition, with accuracy with increasing intensity accuracy increases and, at a moderate intensity, accuracy plateaus. There is evidence that the accuracy rate shows dramatic improvements as expression intensity increases and little further improvement for relatively high-intensity expressions [62,65,66]. However, the accuracy rate might be decreased during peak intensities of emotion since positive expressions and negative expressions look similar at this moment [7,55,67].

For gaze patterns, first, participants fixated more on intense winning faces in the winning task condition and fixated more on intense losing faces in the losing task condition. These results implied that participants were aware of the objective difference between intense winning faces and losing faces. Second, participants gazed more toward the upper facial region (eyes) than the lower region (mouth) to intense and moderate losing faces, and participants were more likely to focus on the lower facial region (mouth) than the upper region (eyes) in winning faces. These results were in accord with previous findings, which also revealed that participants fixated more on the region of the lips for joyful faces and gazed more on the eyes for sad faces [68]. Research has also indicated that the eye region in negative expressions attracts greater fixation and longer viewing times relative to the mouth region [69]. Greater eye constriction contributes to higher intensity ratings of negative emotion in negative faces [70]. The smile is the most common feature of joy [71]; similarly, our findings demonstrated that participants fixated more on the upper region of intense losing faces and the lower region of intense winning faces, implying two critical regions for the recognition of intense facial expressions. This eye movement pattern might reflect a top-down knowledge of what winning and losing faces look like, indicating a goal-driven strategy influences eye gaze behavior during emotional recognition of faces [68]. These results are consistent with the idea that focusing attention on certain diagnostic regions for emotional expressions is beneficial for emotion processing [57,68]. Third, the present study revealed the marginally significant effect of greater fixations on the mouth region of winning faces relative to the eye region when participants were asked to choose a winning face from a pair of facial expressions, and these eye movement patterns were the same for both intense faces and moderate faces. These findings are in line with previous findings [8], which showed that winners are more likely to display more facial muscle activities involving smiling (AU12) and mouth opening (AU27).

Our study also has several potential limitations. First, the intense emotional faces in the present study were selected from the emotional facial expressions with these four facial action units: brow lowering (AU4), smiling (AU12), constriction (AU6), and mouth opening (AU27). Therefore, it seems unlikely that our findings cannot generalize to the full range of spontaneous expressions that occur when people react to a win or a loss. Future studies should enlarge the facial stimulus set and maintain all resulting facial stimuli irrespective of the action units that show up. Second, the moderate emotional faces in the present study were chosen based on the specific valence they expressed. Future studies should also choose any expression that occurs, regardless of the specific valence. Third, previous work found that the arousal of winners was consistently higher than the arousal of losers [7,8]. However, in our experiment, we did not find a significant difference in the arousal-rating scores between intense win faces and intense lose faces. The reason for this might be related to the relatively fewer intense facial images in the present study. Future studies should enlarge the number of facial images to validate the findings of the current research. Fourth, it is important to point out the limitations of the experimental images used in the present study. Facial images in the current experiment were selected from facial expressions of winning and losing in professional tennis matches in real life. Future work may consider examining the differences between intense positive expressions and negative expressions in other situations (e.g., suffering a bereavement, receiving a surprise, etc.). Finally, considering that the force-choice response task is relatively uncommon in the classical facial recognition studies and the limited validity of this method in this research, further investigation is warranted to verify these intriguing findings by using more valid methods or more novel techniques (e.g., ERP techniques with high temporal resolution).

## 5. Conclusions

Above all, the present study adopted a forced-choice task to investigate whether participants could differentiate between winning faces and losing faces produced in professional tennis match scenarios. The results revealed that the accuracy rate of the forced-choice task exceeded the level of chance, and perceivers exhibited differential eye-movement patterns in response to intense winning faces compared with intense losing faces. Overall, these results provided preliminary evidence suggesting that perceivers could perceive the difference between intense positive faces and intense negative faces during peak intensities of emotion in professional tennis matches. A future study might explore other types of intense emotional reactions in real-life situations, such as intense pain and anger, to validate the results of this research.

## Figures and Tables

**Figure 1 behavsci-14-00185-f001:**
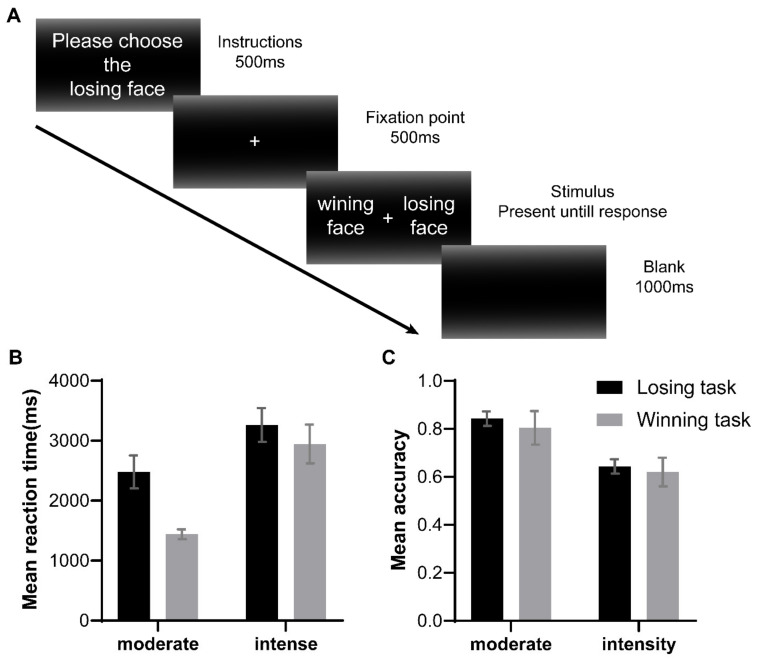
(**A**) An example of an experimental trial in a forced-choice response task. (**B**) Mean reaction times for valence judgment for losing task (black bar) and winning task (gray bar). (**C**) Mean accuracy of valence judgment for losing task (black bar) and winning task (gray bar). Error bars indicate standard errors of the mean.

**Figure 2 behavsci-14-00185-f002:**
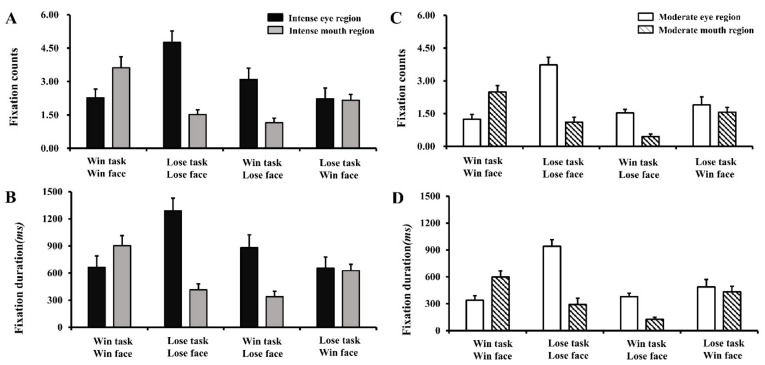
Bar graphs of simple main effect analyses of face type (winning faces vs. losing faces) and task type (choosing the winning face vs. choosing the losing face) on (**A**) average fixation counts for the eye region and the mouth region of intense faces; (**B**) average fixation durations for the eye region and the mouth region of intense faces; (**C**) average fixation counts for the eye region and the mouth region of moderate faces; (**D**) average fixation durations for the eye region and the mouth region of moderate faces. Intense eye region = the eye region of intense faces; intense mouth region = the mouth region of intense faces; moderate eye region = the eye region of moderate faces; moderate mouth region = the mouth region of moderate faces; win task = choosing the winning face from a pair of faces with different valence; lose task = choosing the losing face from a pair of faces with different valence; win face = the gaze behavior toward the winning face; lose face = the gaze behavior toward the losing face. Error bars indicate standard errors of the mean.

## Data Availability

The data presented in this study are available upon request from the corresponding authors.

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
