# Peer review of "Can Perceivers Differentiate Intense Facial Expressions? Eye Movement Patterns"

_behavsci, 2024, doi:10.3390/bs14030185_

Round 1
Reviewer 1 Report
Comments and Suggestions for Authors
While the paper introduces a novel experimental approach named a forced-choice task, we find a discrepancy between the methods employed to measure the characteristics sought by the authors and the experimental techniques used.
The assertion that the presentation of positive images elicits positive emotions in users is acknowledged. However, the effectiveness of simultaneously presenting positive and negative images in evoking emotions remains unclear. Specifically, the validity of emotional valence as an evaluation index is questionable. The concern is not solely about the intensity of the emotions but rather about the suitability of the experimental method employed in this study. The experimental system bears similarities to a gaze search task, raising doubts about the appropriateness of the proposed method for evaluating emotions.
Author Response
Thanks for your valuable comments! According to your suggestions, we would like to modify our expereimental design and improve the quality of our study. We aslo have some points to clarify. First, we employed a forced-choice response task paradigm in the present study to investigate whether participants could discriminate the intense positive emotional expressions from the intense negative emotional expressions. Second, we did not explore whether our experimental materials could induce participants' positive feelings or negative feelings or evaluating emotions.

Reviewer 2 Report
Comments and Suggestions for Authors
In this manuscript the authors test if perceivers can differentiate intense positive and negative facial expressions using behavioural forced choice and eye-tracking. Replicating previous work, participants did not differentiate the valence of the positive and negative expressions when viewed independently, however, they showed above chance differentiation when judging which face in a pair was a winner/loser. Eye-movements to the faces showed distinct movements towards winners vs losers as well. Similar findings (albeit with higher accuracy) were reported for moderate facial expressions.
I found the topic interesting however I have several issues that I found concerning.
Major:
1) Stimuli.
*High intensity images: The high intensity set was curated using a Google image search following keywords in previous work. However, the authors state that they “mainly focused on these facial action units: brow lowering (AU4) and smiling (AU12), constriction (AU6) and mouth opening (AU27) (pg 4 line 153). This statement indicates that the authors “mainly focused” i.e., selected, their stimuli based on some a-priory assumptions and that facial expressions that did not fit these predictions were not chosen. This practice would bias the stimuli set by constricting the full range of natural expressions that occur when people react to a win or a lose. Considering the fact that the set included only 25 intense winners and 25 intense losers (previous work used more than triple than that) potential biases in the current set cannot be disregarded. The authors should enlarge their stimuli set and maintain all resulting stimuli irrespective of the action units that show up.
*Moderate intensity images: This set of images was curated from videos 1000ms after an intense expression was displayed. Here the authors are more explicit about the fact that they specifically chose stimuli that were associated with positive or negative expressions. Thus, the images were not only moderate, they were chosen based on the specific valence they expressed. I’m not sure how this flaw in stimuli selection can be rectified. If the authors want moderate expressions (defined as those occurring 1000ms after an intense expression), they should choose any expression that occurs, not only those that fit their predictions.
2) Results.
*I found it peculiar that the although the valence ratings of individual intense faces replicated previous work (i.e., no difference in valence between intense win and lose face images), the arousal ratings did not replicate previous findings. Specifically, previous work (Aviezer et al 2012, 2015) found that the arousal of winners was consistently higher than the arousal of losers, however, in the current paper, there is no difference in arousal ratings between intense winners and losers (pg 5, line 215). This finding is not discussed, and may reflect the fact that the stimuli selection resulted in a biased set, a concern mentioned earlier.
* The authors conclude (page 7 line 302) that “...participants differentiated the valence of intense facial expressions successfully”. I would tone down this conclusion considering the fact that chance level is 50% and participants were on average accurate on ~60% of the trials. “Above chance” is more appropriate.
3) Eye tracking.
*The authors use the measure of fixation counts and fixation duration which does not seem appropriate. The images were not presented for a fixed duration, rather, they remained on screen until a response was executed. Because the moderate expressions were easily classified, the time they remained on screen was much shorter. Consequently, fewer fixations were made, by default. A better measure would include some proportion from total fixations etc.
*As previous work has shown, intense winners and losers differ objectively in the facial actions they display. It is therefore not surprising that each category attracts eye movements to different parts of the face. What would be interesting is to examine if successful differentiation resulted in different eye movements than failed differentiation. Because around 30-40% of the pairs of intense faces were not differentiated correctly, the authors could compare the eye movements patterns in those cases to the pairs in which differentiation was successful. If no difference is found, we may conclude that the eye movements were unrelated to some explicit successful strategy, but rather reflect differences in the images, as previously noted.
Minor:
*The manuscript needs significant editing to correct English grammar, typos, and style.
*The text in Figure 2 is difficult to read.
* The authors state in the discussion (page 9, line 374) that they investigated “whether participants could explicitly recognize the valence of intense facial expressions”. In fact, this was not tested in the main experiment. Asking participants which face won/lost is not synonymous with explicitly recognizing the valence as positive or negative. For example, both winner and loser may appear equally negative (or positive), but some other cue may give away that one of them lost and the other won (e.g., participants may assume that Novak Djokovic is more likely to win than some anonymous player).
*Considering the above example, did the authors actually examine if the successful differentiations involved more famous, high ranked tennis players?
Comments on the Quality of English LanguageAs noted, the manuscript needs significant editing to correct English grammar, typos, and style.
Author Response
Thanks for your valuable comments! Please see the attachment.

Reviewer 3 Report
Comments and Suggestions for Authors
The Introduction section is perfectly processed. But it is very long. I propose to divide this section into an Introduction and a section Related Work. In the Introduction section, provide only basic information about the research, about your goals from the experiment, and short information about the individual parts of section from your research work. You can include in this moment the Introduction section as complete in the Related Works section (as table historical facts).
Reference 18, 19, 22, 35, 39…. and others are not in text. Please check this and correct. From text is not clear how you rate valence and arousal. Ok, I know that you using 7-likert scale but missing interval of valence and arousal. I premise that you using interval [-1;1]. Using you methods from Russell or another? This missing in your text. Please input this exactly information in to the text.
The pictures are blurry. Please, change your diagrams (charts) as modern. The section Discussion is very long, divide this section to the section Discussion and section Conclusion.
Comments on the Quality of English LanguageQuality of English is not problem.
Author Response
Thanks for your valuable and helpful comments! Please see the attachment.

Round 2
Reviewer 1 Report
Comments and Suggestions for Authors
The primary objective of this research is to investigate the recognition of intense facial expressions. However, certain concerns regarding the methodology and controls have been raised. One notable issue pertains to the image selection method; it remains unclear whether the selection process was entirely random. Were there potential biases in the facial expressions during the selection phase? It is crucial to ascertain if subjects had a role in determining the criteria for selecting images, as this could introduce unintended biases into the study.
Additionally, the 1/2 chance of encountering either face raises questions about the memorization of facial characteristics through repeated comparisons. Was there a discernible difference in correct answer rates between the first and second halves of the study? This information could provide insights into the effectiveness of memorization strategies and their impact on overall performance.
While the conventional method of displaying each face individually allows for a detailed examination of where human attention focuses on intense faces, the comparison with the proposed method needs further clarification. Is there validity in directly comparing the two approaches, and does the unconventional method offer unique insights that the conventional method might overlook?
It is essential to address these concerns and provide a more comprehensive discussion on the validity of the methodology employed in this research.
Reviewer 2 Report
Comments and Suggestions for Authors
The authors have addressed my comments to the best of their ability. The final point that is worth making is that the abstract should indicate that eye movements did not differ for failed or successful trials.
Aside from that i have no further requests.
Comments on the Quality of English LanguageThe quality of English has improved.
Round 3
Reviewer 1 Report
Comments and Suggestions for Authors
I have ensured that the author has addressed my comments to the best.